# Flight Attitude Estimation with Radar for Remote Sensing Applications

**DOI:** 10.3390/s24154905

**Published:** 2024-07-29

**Authors:** Christoph Weber, Marius Eggert, Thomas Udelhoven

**Affiliations:** 1Engineering, Computer Science and Economics, TH Bingen University of Applied Sciences, 55411 Bingen am Rhein, Germany; 2Faculty of Design Computer Science Media, RheinMain University of Applied Sciences, 65197 Wiesbaden, Germany; marius.eggert@hs-rm.de; 3Environmental Remote Sensing & Geoinformatics Department, University of Trier, 54286 Trier, Germany; udelhove@uni-trier.de

**Keywords:** radar sensor, flight attitude estimation, drone, UAV, remote sensing

## Abstract

Unmanned aerial vehicles (UAVs) and radar technology have benefitted from breakthroughs in recent decades. Both technologies have found applications independently of each other, but together, they also unlock new possibilities, especially for remote sensing applications. One of the key factors for a remote sensing system is the estimation of the flight attitude. Despite the advancements, accurate attitude estimation remains a significant challenge, particularly due to the limitations of a conventional Inertial Measurement Unit (IMU). Because these sensors may suffer from issues such as drifting, additional effort is required to obtain a stable attitude. Against that background, this study introduces a novel methodology for making an attitude estimation using radar data. Herein, we present a drone measurement system and detail its calculation process. We also demonstrate our results using three flight scenarios and outline the limitations of the approach. The results show that the roll and pitch angles can be calculated using the radar data, and we conclude that the findings of this research will help to improve the flight attitude estimation of remote sensing flights with a radar sensor.

## 1. Introduction

In recent decades, drone or unmanned aerial vehicle (UAV) technology and radar systems have seen innovative breakthroughs, which have opened up applications that were previously impossible [1,2]. Not only together but also independently, both technologies have been the focus of significant research and applications. The various fields of drone applications extend from disaster management [3,4] to agriculture [5,6], forest inspection [7,8], delivery [9,10] and construction site projects [11,12]. In order to identify drones, especially in restricted airspaces, motion capture systems are used to detect and track UAVs [13,14,15].

Radar systems have had a particularly strong impact in the automotive sector [16], where imaging radar sensors are used, formerly with two-dimensional measurement systems such as the ARS-408 [17] and today with three-dimensional systems like the ARS-508 [18]. These are utilized for advanced driver assistance systems or autonomous driving functionalities such as adaptive speed control [19], emergency brake systems and lane change assistance [16,17]. In addition to automotive applications, radar is also used to detect and localize drones [20,21] and as a synthetic aperture radar (SAR) carried by drones, airplanes, or satellites [22]. Moreover, SAR is often used in combination with satellites for large-scale or Earth observation applications such as sea ice classification [23,24], hurricane observation [25,26], or global biomass mapping [27,28].

In comparison with satellites and airplanes, drones have some advantages such as higher accuracy, flying on demand and lower costs [29,30,31,32]. The combination of drones and imaging radar systems has unlocked new opportunities for remote sensing applications such as crop growth monitoring [33,34], snow depth measurement [35,36], soil moisture mapping [37,38], landmine detection [39,40] and obstacle avoidance [41,42]. Particularly for applications that are intended to collect remote sensing data, an accurate flight attitude estimation of the drone and the remote sensing system is important [43].

In order to fly and navigate to a destination, drones need a flight attitude and position estimation. This estimation has a considerable influence on the data quality of the remote sensing system [44,45]. For long-term scientific studies or ongoing monitoring, a consistent attitude and position enable a reliable comparison over time [46,47]. Then, the remote sensing data collection is comparable and allows trend analysis, change detection and forecasts [31]. For mapping and surveying applications, a correct attitude and position allow for the creation of high-resolution maps, data fusion or multi-sensor mapping [48,49,50]. Often, mapping applications require the data to be linked to a geographic information system (GIS) for accurate georeferencing [51,52].

Flight attitude and position estimation techniques include the Inertial Measurement Unit (IMU), Global Navigation Satellite System (GNSS), ultrasonic distance sensors, barometric pressure sensors and vision-based systems [53,54,55]. To reach an acceptable accuracy, the sensors values are often merged in a sensor fusion process such as the extended Kalman filter (EFK) [56]. Furthermore, technologies such as Real-Time Kinematic positioning (RTK) or the Differential Global Positioning System (DGPS) are used to improve the positioning [57,58]. However, both systems require a base station, which entails great effort and high costs. Other authors use artificial intelligence [5,53,59] or a special controller such as the Linear Quadratic Regulator [60,61] to improve attitude and position estimations. However, the IMU is a crucial sensor for the stability of the drone and, thus, for the quality of the remote sensing data [43,62]. To ensure the availability of the IMU data in the event of a total failure, a redundant IMU can be used. Nonetheless, long-term drift, signal noise, offset error and temperature sensitivity stand as problems [63,64,65].

This research was aimed at uncovering the potential of drones with radar sensors in attitude estimation for remote sensing applications. We demonstrate that this approach fills a gap in the attitude estimation of remote sensing systems, since a drone with a radar sensor can achieve higher accuracy without requiring any additional hardware effort. Furthermore, we show that this attitude estimation can be used as a backup solution or for sensor fusion.

The main goal when applying our methodological approach was to make an attitude estimation based on the radar sensor data. To achieve this, we used a drone system equipped with an ARS-548 radar sensor [18,66] and other sensors. We applied common mathematical methods such as matrix equations and the Moore–Penrose inverse to support our real-time processing. Ultimately, we present our results when comparing three flight measurements with the drone system and highlight the advantages and drawbacks of the proposed method.

## 2. Materials and Methods

### 2.1. Drone and Radar Sensor

For this research, an ARS-548 radar sensor from the manufacturer Continental was used. It has an operating frequency of 77 GHz and was originally developed for use in cars and trucks. Therefore, the radar runs on customized software with respect to the vehicle and its functions. It is often mounted in the front or rear bumper. To use this radar for other applications, a special software was designed by the manufacturer that gives access to all radar data.

In comparison to the previous version, the ARS-408, this new radar sensor has a three-dimensional measurement. Therefore, this radar provides three-dimensional data with a single measurement. The axes from the radar, which are annotated with a subscripted R, are defined as follows:X_R_: The distance measured from the radar sensor straight ahead;Y_R_: The azimuth angle (left and right);Z_R_: The elevation angle (up and down).

When used in vehicles, the axes’ description complies with the normal cartesian coordinate system defined by *X*, *Y* and *Z*. For applications on a drone, this depends on the mounting of the radar sensor. Figure 1a shows the drone with the radar rotated by 90° so that it is pointed at the ground, which corresponds to the setup we used for our tests. It is the preferred setup for remote sensing applications as it scans a large area on the ground.

Figure 1b shows the front view of the drone with the IMU and radar sensor attached underneath. The radar sensor is mounted between the landing skid and the IMU on the back side of the radar.

If the radar is installed as shown in Figure 1a, the coordinate system of the radar is different from the drone and the IMU coordinate system shown in Figure 2.

The following is defined using Figure 2:X = −X_IMU_ = Z_R_: Area in front of and behind the drone;Y = Y_IMU_ = Y_R_: Area to the left and right sides of the drone;Z = −Z_IMU_ = −X_R_: Flight height above the ground (AGL) or distance from the ground.

For data processing, all coordinate systems need to be the same. Therefore, it is necessary to rotate around the *Y*-axis. The drone’s coordinate system is defined as the target coordinate system, and the IMU and radar are rotated by −180° and 90°, respectively.

In addition to the X_R_-, Y_R_- and Z_R_-axes, the radar sensor outputs the radar cross-section (RCS) data and the range rate as described in Table 1. The RCS value is the reflection strength of the detected object in decibel square meters (dBm^2^). The range rate is the one-dimensional velocity of the object in relation to the radar sensor. Furthermore, the radar sensor outputs the redundant data range, azimuth and elevation.

A single measurement of the radar is called a radar image. A radar image consists of radar detection points. The number of detection points depends on the reflection characteristics of the detection area and can range from a few up to 250 points per image. The arrangement of the points in the image is random and depends on the detection area as well. The radar sensor ARS-548 outputs a radar image every 50 milliseconds. Unlike camera images, individual radar images are difficult to interpret, and scenarios can look different from image to image. A radar measurement consists of many radar images.

In our setup, the radar sensor is carried by a DJI Matrice M300 drone (DJI, Shenzhen, China). It is equipped with additional sensors such as an IMU, GNSS and RGB camera. All sensors, including the sensors of the drone, are recorded for evaluation. The IMU is a Bosch BNO055 and is mainly used to compare the results of the radar data. Therefore, the IMU is mounted directly on the backside of the radar sensor. The advantage of this IMU sensor is a fusion algorithm that can output the attitude as quaternions or Euler angles. For this, three integrated sensors—an accelerometer, gyroscope and magnetometer—are used. To control the recording system of all sensors, an Nvidia Jetson Nano is installed. To connect it with the radar sensor, a media converter (MC 100BASE-T1 BCM from Technical Engineering, Munich, Germany) is used. A 1 terabyte USB SSD is utilized to store the recordings. The result of a recording is a db3 file, which holds an SQLite database with Robot Operating System (ROS) messages from all sensors.

### 2.2. Recordings

To evaluate our method, three recordings were taken. All measurements used the flight setup shown in Figure 1.

The altitude of the first flight was 30 m, and the direction was north–south. Overall, the recording was 190 s long. The actual flight started at 70 s and ended at 146 s, with take-off and landing processes taking place before and after. There were two recording sections, the outbound and the return flight, which were divided by a 180° yaw turn. The terrain was a flat, an open field with grass and no obstacles apart from the four radar corners.

The second flight had an altitude of 35 m and a west–east direction. The recording was 11 min long and contained eight straight flights. The drone did not make a yaw turn but flew backwards. Only the last straight flight contained a 180° yaw turn. The terrain was an open field with one tall high-voltage pylon and two small pylons. The terrain sloped down toward the east.

During the third flight, the altitude was 35 m, and the area was overflown with four straight flights in the south–southwest direction. The recording was 22 min long. The four main flights were connected by 180° turns, which were subdivided into a 90° turn followed by a short straight flight and, then, another 90° turn. This ensured that the large area was scanned efficiently with a defined overlap. In total, an area of around 200,000 m^2^ was scanned. The area was an industrial plant with buildings, containers, conveyor belts, bulk material dumps, woodland, sand and water areas.

### 2.3. Calculation Process

The whole process has five main steps with several sub-steps, as Figure 3 shows. In this chapter, we focus on the three processing steps.

In the pre-processing stage, the data will be read from the database file and saved in three structures for further processing:gRadar: Data from the radar sensor;gJet: Sensor data attached to the Nvidia Jetson Nano;gUav: Sensor data from the DJI Matrice M300 drone.

In the next step, the IMU data need to be rotated to fit the orientation of the drone. For this, a rotation matrix (shown in Equation (1)) is used, which rotates the Euler angles (eul) by −180° around the *Y*-axis.
(1)$${eul}_{rot}=\left[\begin{array}{ccc} -1 & 0 & -0\\ 0 & 1 & 0\\ 0 & 0 & -1 \end{array}\right]\cdot eul\;\mathrm{with}\;eul=\left[\begin{array}{c} roll\\ pitch\\ yaw \end{array}\right]$$

The calculation process is designed to determine the attitude by analyzing the radar data, focusing on identifying the ground plane and filtering out irrelevant data points. As such, the ground plane has the most reflections in the radar data, which can be visualized with a three-dimensional histogram. Figure 4 shows the first recording with all radar images and the distance *X*_R_ to the ground, as well as the count of how often this distance occurs. It shows that most counts are in the 30 m to 33 m bars, which correspond to the original flight altitude of 30 m. At the beginning and end of the histogram, the distance decreases due to the take-off and landing of the drone.

Before calculating the attitude of the radar and, thus, the drone, it is necessary to filter the data. In particular, radar data far from the ground plane make the calculation process instable. To overcome this problem, the *X*_R_-axis is filtered by the flight altitude results from the histogram. For this, a hysteresis is used. With a hysteresis of 5 m, all radar detections lower than 25 m and higher than 35 m are excluded. With this filtering, we have no attitude estimation at take-off or at landing. An alternative filtering process is to exclude outlier data points. The mean absolute deviation method has proven to be robust and suitable.

After the pre-processing is completed, the actual calculation can be performed. The complete process will be performed for each radar image. The calculation process can theoretically be solved with three data points per image, but this can lead to instabilities. Therefore, data processing should be aborted if too few data points are available per radar image. Our tests suggest that stable results can be achieved with at least 10 points per image.

The filtered radar data are then used to calculate the orientation of a plane. Since we have a largely overfitted dataset, we use a least-squares method with the plane equation ax+by+c=z. Using the given x, y and z values from the radar data, the coefficients a, b and c must be determined. Therefore, we set up the matrix equation seen in Equation (2) and insert the coordinates $x_{i}$, $y_{i}$ and $z_{i}$ of each measured point $p_{i}$.
(2)$$A\left[\begin{array}{c} a\\ b\\ c \end{array}\right]=B,\;\mathrm{with}\;A=\left[\begin{array}{ccc} x_{1} & y_{1} & 1\\ x_{2} & y_{2} & 1\\ \vdots & \vdots & \vdots \\ x_{n} & y_{n} & 1 \end{array}\right]\;\mathrm{and}\;B=\left[\begin{array}{c} z_{1}\\ z_{2}\\ \vdots \\ z_{n} \end{array}\right]$$

Then, we calculate the Moore–Penrose pseudo-inverse $A^{+}={\left(A^{T}A\right)}^{-1}A^{T}$ and apply it to B, as shown in Equation (3), so values for a, b and c can be calculated.
(3)$$\left[\begin{array}{c} a\\ b\\ c \end{array}\right]=A^{+}B$$

To determine the orientation of the plane, we first calculate three points located on the plane by using the planes’ equation (see Equation (4)).
(4)z(x,y)=a∗x+b∗y+c

To do this, we use the (x, y) values $\left(0,\;0\right)$, $\left(0,\;1\right)$ and $\left(1,\;0\right)$, as shown in Equation (5), and pinpoint the three points (Equation (6)) on the plane.
(5)$$z_{1}=a*0+b*0+c\;z_{2}=a*0+b*1+c\;z_{3}=a*1+b*0+c$$
(6)$$p_{1=}\left[\begin{array}{c} 0\\ 0\\ z_{1} \end{array}\right];p_{2=}\left[\begin{array}{c} 0\\ 1\\ z_{2} \end{array}\right];p_{3=}\left[\begin{array}{c} 1\\ 0\\ z_{3} \end{array}\right]$$

Then, two linear independent vectors are calculated by subtracting the two outer points ($p_{2}\;$ and $p_{3}$) from $p_{1}$, which is located at the origin (Equation (7)). Using the cross product of these two vectors’ results in a perpendicular vector $v_{p}$ describing the planes’ orientation relative to the drone (Equation (8)).
(7)$$v_{1}=\left[\begin{array}{c} 0\\ 1\\ z_{2} \end{array}\right]-\left[\begin{array}{c} 0\\ 0\\ z_{1} \end{array}\right];v_{2}=\left[\begin{array}{c} 1\\ 0\\ z_{3} \end{array}\right]-\left[\begin{array}{c} 0\\ 0\\ z_{1} \end{array}\right]$$
(8)$$v_{p}=\left[\begin{array}{c} v_{p1}\\ v_{p2}\\ v_{p3} \end{array}\right]=v_{2}\times v_{1}$$

Figure 5 shows the radar detection points as blue points. With the described calculation process, the black plane as well as the perpendicular vector in magenta can be determined. For a better understanding, the normal cartesian coordinate system vectors in red, blue and green are shown.

The roll and pitch angles in degrees can then be reconstructed from $v_{p}$ using trigonometry, as shown in Equation (9).
(9)$$roll=\frac{180}{\pi }*\tan^{-1}\frac{v_{p1}}{v_{p3}}\;pitch=\frac{180}{\pi }*\tan^{-1}\frac{v_{p2}}{v_{p3}}$$

This concludes the process that is necessary for each radar image. It is not possible to calculate the yaw angle. In the post-process, the calculated angles need to be rotated around the *Y*-axis by 90° (Equation (10)).
(10)$${eul}_{rot}=\left[\begin{array}{ccc} 0 & 0 & 1\\ 0 & 1 & 0\\ -1 & 0 & 0 \end{array}\right]\cdot eul\;\mathrm{with}\;eul=\left[\begin{array}{c} roll\\ pitch\\ 0 \end{array}\right]$$

Next, the roll and pitch angles can be limited to a certain value range. For normal flight scenarios, a boundary of ±15° is appropriate. For turbulent flights, the data may also be smoothed using a moving median filter. This completes the entire process.

## 3. Results

The three recordings were evaluated by comparing the calculated radar data with the IMU mounted on the backside of the radar sensor. Since the recording of the first flight contained the fewest obstacles, it was used to demonstrate the general applicability of our method. Subsequently, these results were compared with the second and third flight.

### 3.1. Open Field Flight

Figure 6 shows the roll and pitch angles of the complete first flight reconstructed from the radar (green) in comparison with the IMU (red). The calculation process is set up with a minimum of 10 detections per radar image and outlier deletion. For radar images with fewer than 10 detections, an angle of 0° is assumed.

Until about 50 s, there is no angle from the radar, as the number of radar detections is too low while the drone stands on the ground. Between 50 and 60 s is the take-off procedure, where the number of radar detections fluctuates greatly, so the angles often fall back to 0°. The same applies to the end of the measurement, where the drone is landing. Between take-off and landing, the radar angles follow the IMU values, especially during the two main flight parts at 80–100 s and 110–135 s but also at extrema such as 70 s and 105 s. A 180° yaw curve separating the two main flight sections occurs at 105 s. The roll angle is lower than zero before the turn and higher than zero afterward, which is caused by a side wind. The pitch angle of both main flight sections is less than zero because the drone is slightly tilted to move forward. In comparison to the pitch angle, the roll angle is significantly smoother. This is because the roll angle is calculated by the large Y_R_ area of the radar. In contrast, the Z_R_ area is much smaller, and thus, the calculated pitch angle is noisier. In particular, between 50 and 80 s, the angle is very instable.

Figure 7 shows the number of radar detections per radar image. In combination with Figure 6, it can be assumed that a minimum of about 50 radar detections per radar image is sufficient for good and stable angles, whereby the roll angle is stable earlier than the pitch angle.

Most of the time during the main flight, a radar image has about 200 radar detections, and therefore, the equation is significantly overdetermined. To reduce the processing time, random radar detections were excluded. Figure 8 shows how the angle changes depending on the number of detections per radar image.

Because the radar detections are randomly selected for the calculation, the figure looks different for each instance of processing. The radar image uses a total of 243 detections, and the roll angle in red is already relatively stable after 75 detections. After that point, the angle deviates within ±0.2°. The pitch angle in green shows a ±1° deviation after 125 detections. When they are directly compared, the roll angle is significantly smoother than the pitch angle.

The total processing time for the complete flight and, thus, 2808 radar images and 297,446 detection points is about 7.4 s. The average processing time per radar image is 2.6 milliseconds. On a computer with an Intel i7 CPU, reducing the number of detections for this flight to a maximum of 75 detections has no effect on the processing time. However, this aspect may be more relevant for less powerful systems.

To reduce the noise, especially for the pitch angle, filtering with a moving median is suitable. Figure 9 shows the complete measurement with a filter window size of 10.

The take-off procedure at 50–65 s improves the roll angle. This shows that the median filtered radar data point is not at 0°, as would be expected based on Figure 6, but between −3° and −5°. The main flight parts do not improve significantly with the filter. The pitch angle improves throughout the flight, not only during the noisy take-off part but also during the main flight.

Figure 10 shows the difference between the IMU and the filtered radar sensor values, where the IMU data are subtracted from the filtered radar data.

Overall, the difference in the roll angle is small. In the first main flight part, the angle converges from −2° to zero. The difference is about 2° for the second flight part. The pitch angle is generally noisier than the roll angle and has some peaks. The differences are approximately −1.6° for the first main flight section and 0° for the second.

Next, the root mean square error (RMSE) is calculated for the following three scopes:IMU data—Radar data.IMU data—Filtered radar data.IMU data—Filtered radar data, only for the main flight parts.

The results are listed in Table 2.

The RMSE shows that filtering the radar data has a negligible influence on the roll angle. The pitch angle has a slight improvement of 0.1°. For the main flight parts, both angles show an improvement. Compared to the roll angle, the pitch angle is significantly improved and achieves better results than the roll angle. The RMSE for the main flight parts shows a very good result.

To further improve the roll and pitch angles, the data of the radar sensor and IMU sensor could be used together. Averaging the values is common; however, it does not bring any improvement for our results as the data are too similar. Furthermore, this approach does not consider the special behavior of the radar, such as during take-off and landing. To improve the angle, a more complex algorithm such as a fusion algorithm is necessary.

### 3.2. High-Voltage Pylon Flight

For the second flight, the same setup and configuration are used as for the first flight. Figure 11 shows the reconstructed, moving-median filtered angles of the radar sensor in green, and the IMU values are shown in red.

Shortly before 100 s, the roll angle deviates from the IMU value. During this time, a tall power pylon is on the right side of the radar, which leads to a miscalculation. The smaller pylons do not disrupt the results of the proposed method. At the end of the main recording, at about 610 s, the tall pylon is on the left side of the radar. Because the drone is already in the landing procedure, this part is also noisy. During the rest of the measurement, the values of the IMU and radar correlate. The roll angle shows a permanent offset of about 2.5°. Based on the peaks of the measurement, the flights can be visually divided into the eight main flight parts.

The processing time for the record is 164 s, with a total of 13,232 radar images and a total of 1,727,032 detection points. The average processing time is 12.4 milliseconds per radar image. The processing time is the same as when the radar detections are reduced to a maximum of 75 per radar image.

Table 3 shows the RMSE of the flight with the three scopes, as already shown for the previous flight (compare with Table 2).

The pitch angle shows better results than the roll because of the roll angle offset. In this flight, the roll angle deteriorates due to the filtering. Apart from the offset, the results are comparable with those of the first flight.

### 3.3. Industrial Plant Flight

During the industrial plant flight, the IMU values show a strong drift, so the data are not a suitable comparison in our research. To evaluate our results, we instead use the data from the DJI drone. The reconstructed angles from the radar sensor and the DJI drone are shown in Figure 12.

The data in Figure 12 are filtered with a moving median and a window size of 20. The roll angle of the radar follows the angle of the DJI drone. At about 350 s, the angles differ because of a forest on the right and a flat sand and water area on the left side. Due to a turbulent flight, the pitch angle is very unstable; this applies to the DJI angles as well as the radar angles. Both angles generally correlate, but at around 1000 s, the radar angles are extremely noisy. The smooth flight part at around 800–900 s is notable. During this time, the drone flies completely over water. Figure 13 shows that this area has significantly fewer radar detections per radar image than all others. Despite the low number of detections, the resulting angle in this area is better than in the rest of the flight.

The processing time for this measurement is 228 s for a total of 18,132 radar images and 1,710,237 points. Thus, the processing time per image is about 12.5 milliseconds on average. As for the first and second flights, reducing the radar detections to a maximum of 75 per radar image has no effect on the processing time.

Table 4 shows the RMSE of the last flight.

As for the first flight, the roll angle shows a better result than the pitch. The filtered radar data show a slight improvement. Once again, the main flight parts show the best results. Compared to the other two, this flight displays the worst results. Not only is the RMSE high but also the IMU reveals problems, and its data are very noisy.

For this flight, the combination of IMU and a radar sensor can improve the angles. Figure 14 shows the differences between the DJI drone angles and the radar as well as between the DJI drone and the IMU.

At the start, the IMU shows slightly better results than the radar, but after the drift becomes stronger, the radar shows better results. Without the angles from the radar data, the complete flight is not suitable for later evaluation. It is also much easier to recognize the drift of the IMU with the radar data. However, to use the improved angle, a fusion algorithm is necessary that automatically combines the IMU and radar data.

## 4. Discussion and Further Challenges

In this research, a methodological approach for calculating the flight attitude of a remote sensing system from radar sensor data was presented. The process to calculate the angles is organized in five main steps with various sub-steps. This process can be run automatically and only needs adjustment if the mounting of the sensors is changed. The calculated roll and pitch angles correlated with the angles from the IMU sensor for all three evaluated recordings.

In the first flight, the RMSEs for the main flight parts were 1.5° for the roll angle and 1.4° for the pitch angle. The RMSEs for the complete flight were higher at 1.6° and 2.5°. Both results showed an overall small deviation. The second flight showed similar RMSEs with 2.5° for the roll and 1.4° for the pitch angle in the main flight parts. It was notable that the pitch angle showed better results than the roll. The last flight was turbulent, and the IMU had strong drift, so it was unsuitable for comparison. Instead, we used the IMU data from the DJI drone, with RMSE results of 5.1° and 7.8°. These values were considerably worse than during the first flights, but the results may be acceptable depending on the specific application. Regarding the errors of the attached IMU, in particular, which would have made the measurements completely unusable, our approach provided a significant improvement. In addition, the inclusion of our results enabled the identification of a clear IMU drift. As this demonstrates, our approach can be used to detect IMU drifts and can serve as a backup solution for evaluating recordings or performing an emergency landing. We assume that a fusion algorithm can combine the sensors and derive the best from each. This will eliminate the need for manual monitoring.

The average runtime for a single calculation on a normal desktop computer was from 2.5 ms to 12.5 ms, depending on the data size. While this can be considered sufficiently fast for real-time calculations, a test of the specific platform available on the particular drone system needs to be performed. Limiting the maximum number of radar detections per calculation had no effect on the runtime during our evaluations. However, it is possible that it had a significant effect on less powerful hardware.

One of the further challenges is to reconstruct the yaw angle. One approach would be to identify objects in the radar image and observe one object to calculate the rotation from one radar image to the next. This way, it would be possible to derive a relative yaw angle. To ascertain an absolute flight attitude, this approach requires the current yaw angle on one occasion, which can be handled in the post-processing or before the flight.

Another challenge is to make the process even more robust against disturbance objects such as large power pylons. For this, separation between the ground and the object improves the results, as the authors of [43] showed. In this way, objects will be excluded from the calculation process.

As we had too few detections, no estimation could be performed during take-off and landing. For this, a radar image with few detections can be compiled with the following images, so that rather than using one radar image for the processing, e.g., five radar images are used together. In this way, an emergency landing using only the radar data becomes possible.

## 5. Conclusions

For remote sensing applications with a radar sensor, our presented method allows the angles, and thus the data quality, to be significantly improved without additional hardware effort. In combination with an IMU, the measuring system provides redundant flight attitude estimation for the roll and pitch angles. Even when the IMU exhibits drift or other measurement inaccuracies, the radar data can still be used. In the event of a worst-case scenario where the IMU fails completely, the radar can be used as a fallback system. However, due to the limitations described, we assume that accurate flight attitude is only possible with a sensor fusion algorithm.

## Figures and Tables

**Figure 1 sensors-24-04905-f001:**
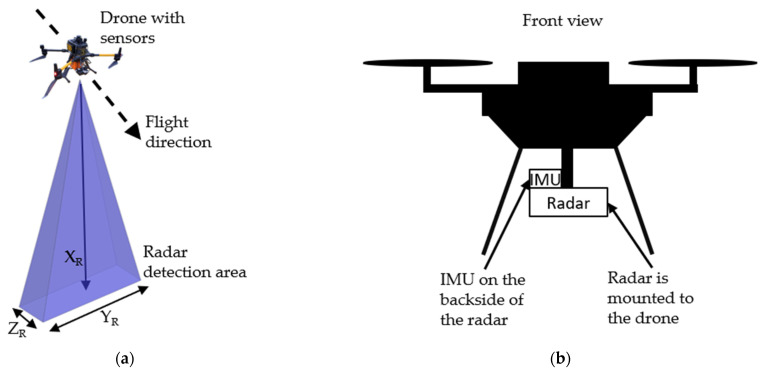
(**a**) Field of view of the radar sensor mounted on a drone including axis description from the radar. (**b**) Radar sensor and IMU mounted on the drone.

**Figure 2 sensors-24-04905-f002:**
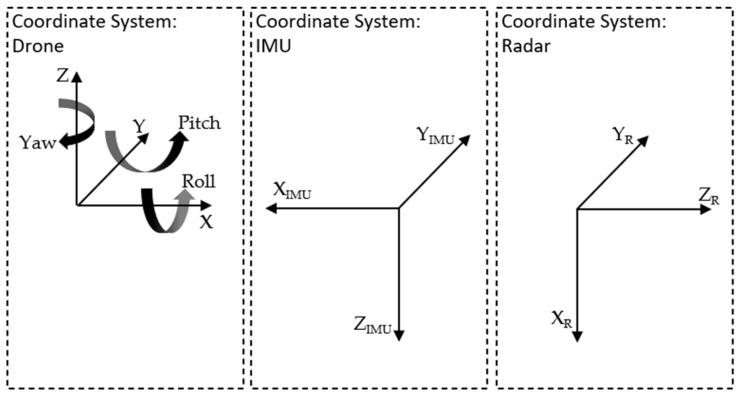
Different coordinate systems of the drone, IMU and radar sensor.

**Figure 3 sensors-24-04905-f003:**
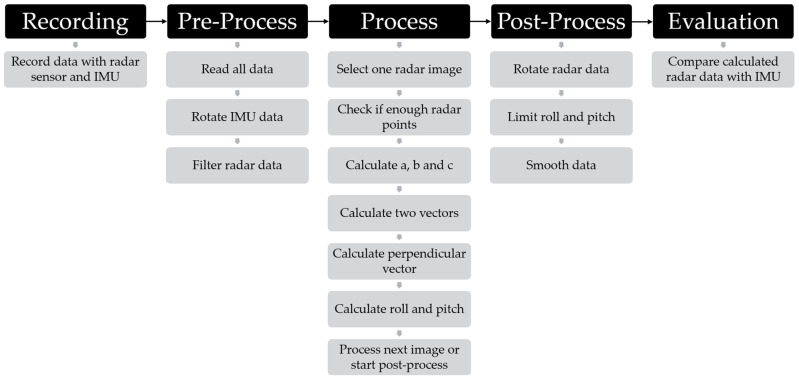
Complete process from recording the data with the drone to evaluating the calculated data in comparison with the IMU data.

**Figure 4 sensors-24-04905-f004:**
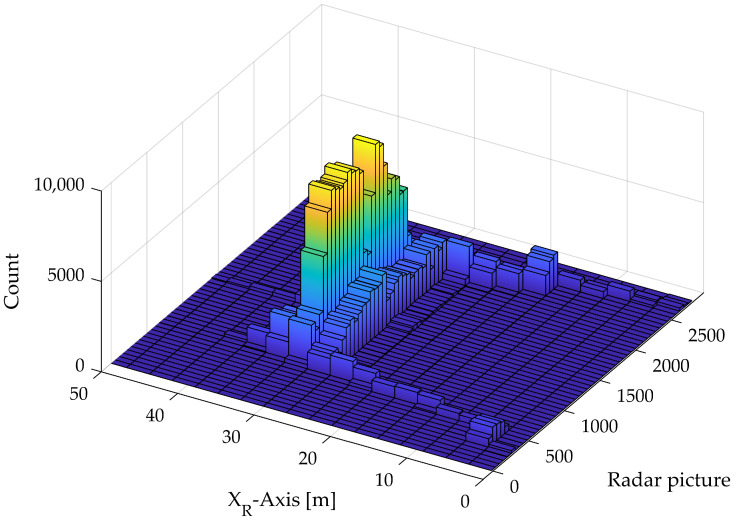
Histogram of a complete flight to identify the ground plane by counting the distance *X*_R_ for every radar image.

**Figure 5 sensors-24-04905-f005:**
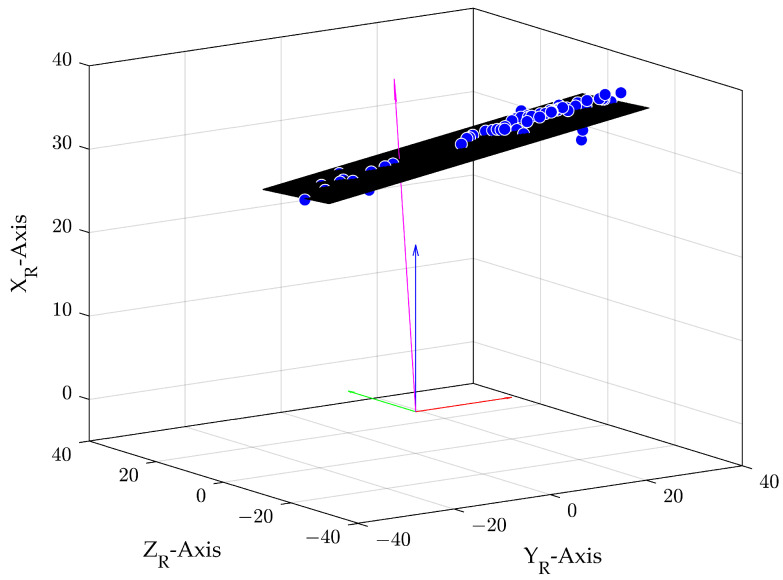
Plane in black with perpendicular vector (magenta) of the calculation process. The blue points are the initial radar detections, and, in red, green and blue, the vectors of the cartesian coordinate system are indicated.

**Figure 6 sensors-24-04905-f006:**
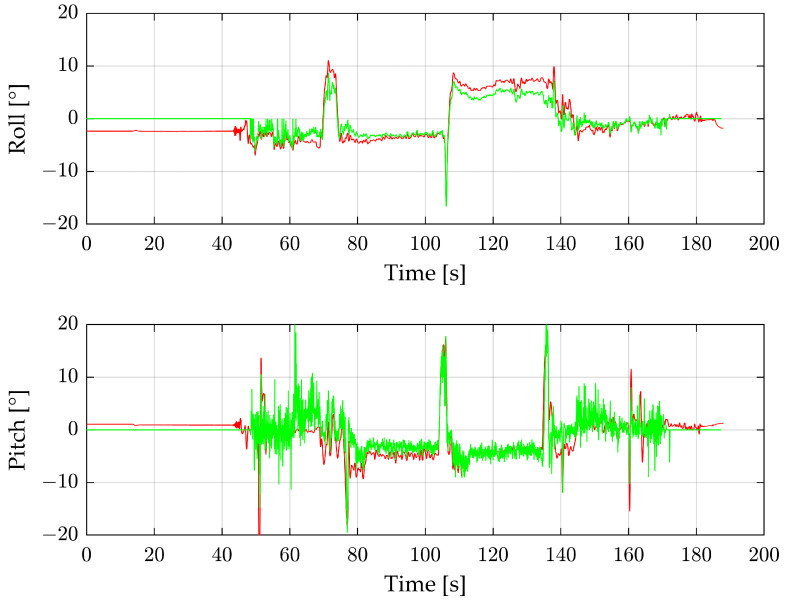
Roll and pitch angles of the complete first flight. Green shows the reconstructed values from the radar sensor, and red the IMU data.

**Figure 7 sensors-24-04905-f007:**
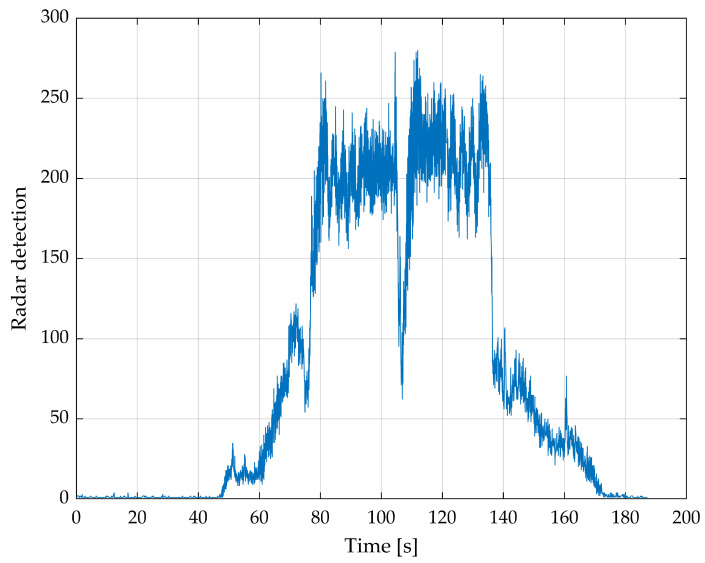
Number of radar detections per radar image of the complete first flight.

**Figure 8 sensors-24-04905-f008:**
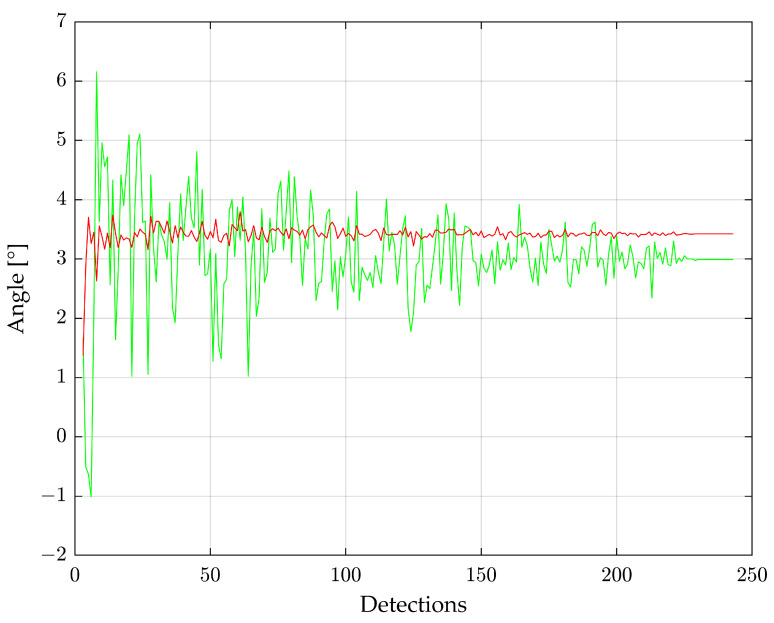
Change in roll (red) and pitch (green) angle as a function of reduction in the detection points for a radar image.

**Figure 9 sensors-24-04905-f009:**
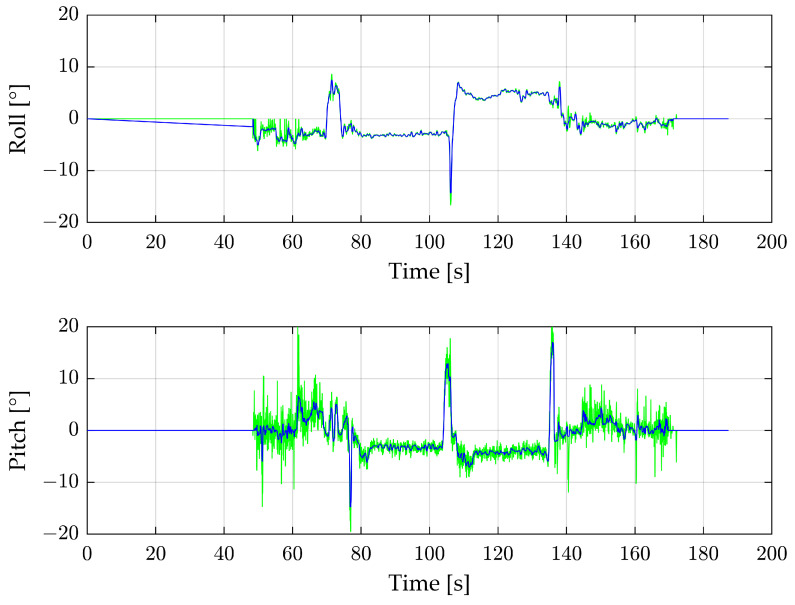
Complete first flight with roll and pitch angles from the radar sensor in green and the moving median filtered signal in blue.

**Figure 10 sensors-24-04905-f010:**
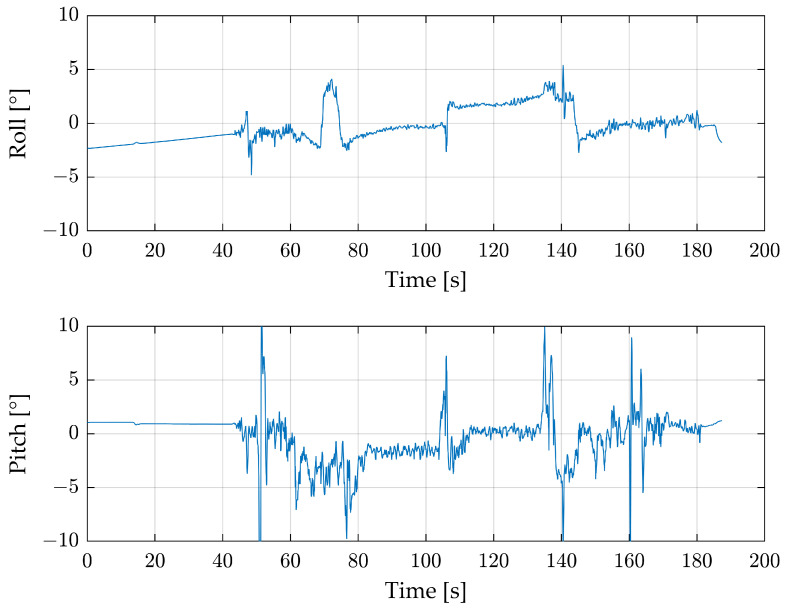
Difference between the IMU and the filtered radar sensor values for the first flight. The IMU data were subtracted from the filtered radar data.

**Figure 11 sensors-24-04905-f011:**
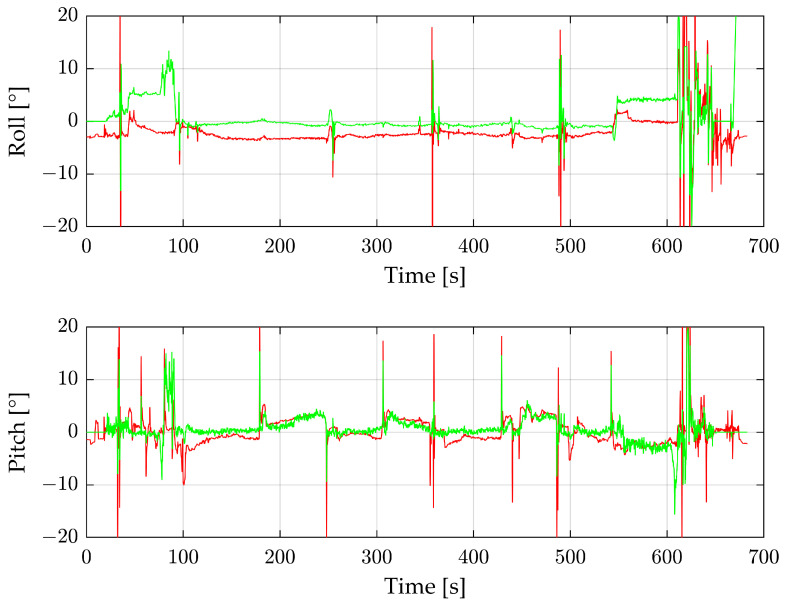
Roll and pitch angles of the complete second flight. Green shows the reconstructed filtered values from the radar sensor, and red the IMU data.

**Figure 12 sensors-24-04905-f012:**
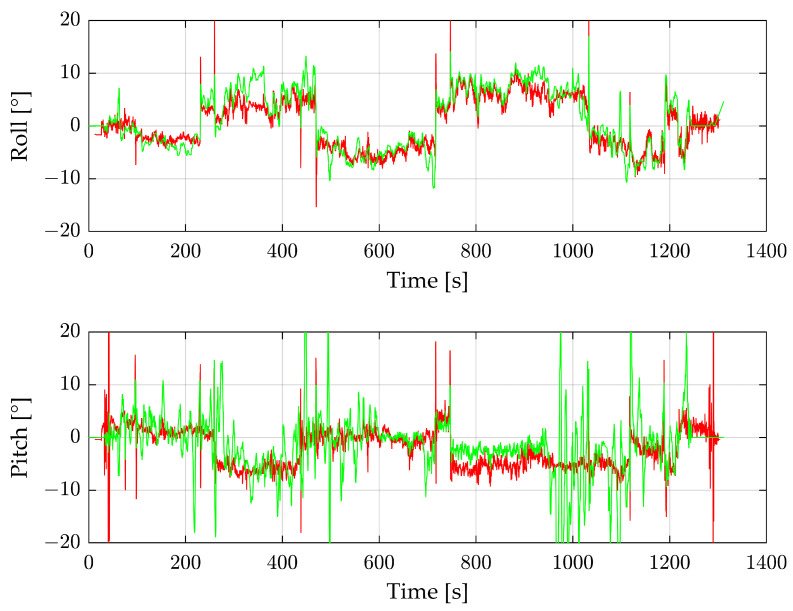
Roll and pitch angles of the complete third flight. Green shows the reconstructed values from the radar sensor, and red the IMU data from the DJI drone.

**Figure 13 sensors-24-04905-f013:**
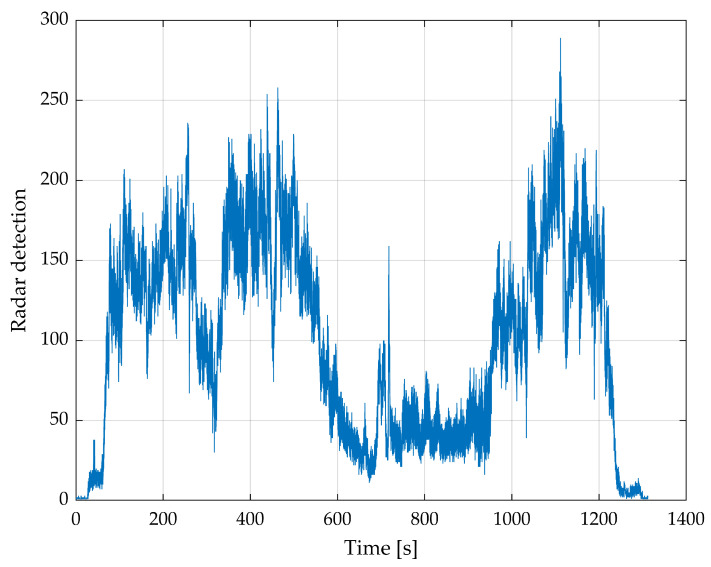
Count of radar detections per radar image of the complete third flight.

**Figure 14 sensors-24-04905-f014:**
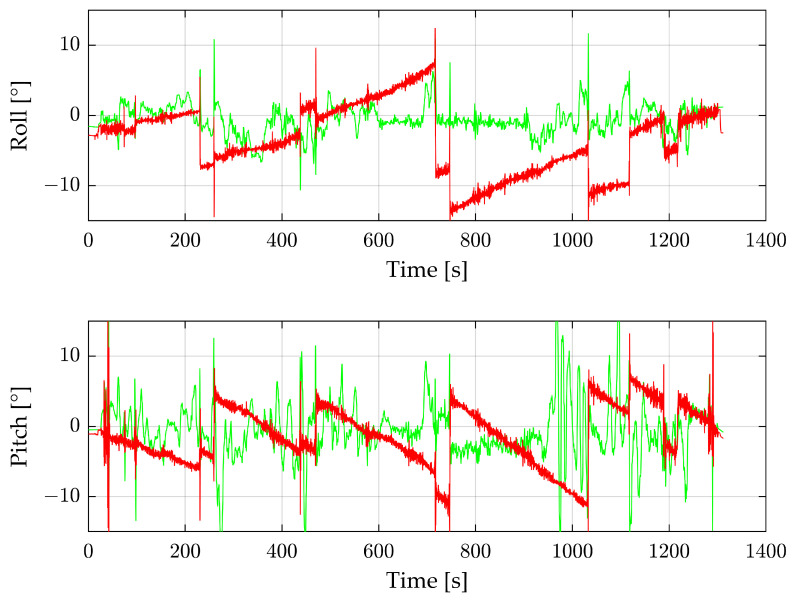
Difference between the DJI drone’s IMU and the filtered radar sensor value in green, and the difference between the DJI drone’s IMU and the IMU Bosch BNO055 in red, both for the third flight.

**Table 1 sensors-24-04905-t001:** Description of the radar sensor data output for single-object detection.

Data	Description	Unit
X_R_	Cartesian coordinate system—*X*	m
Y_R_	Cartesian coordinate system—*Y*	m
Z_R_	Cartesian coordinate system—*Z*	m
Range	Spherical coordinate system—r	m
Azimuth	Spherical coordinate system—φ	rad
Elevation	Spherical coordinate system—θ	rad
Radar cross-section	Detection strength	dBm^2^
Range Rate	Detection velocity	m/s

**Table 2 sensors-24-04905-t002:** Root mean square error (RMSE) of the angles’ roll and pitch of the first flight.

Angle	1. RMSE IMU—Radar	2. RMSE IMU—Filtered Radar	3. RMSE IMU—Filtered Radar Main Flight Parts
Roll	1.6°	1.6°	1.5°
Pitch	2.5°	2.4°	1.4°

**Table 3 sensors-24-04905-t003:** Root mean square error (RMSE) of the angles’ roll and pitch of the second flight.

Angle	1. RMSE IMU—Radar	2. RMSE IMU—Filtered Radar	3. RMSE IMU—Filtered Radar Main Flight Parts
Roll	3.3°	3.7°	2.5°
Pitch	2.8°	2.5°	1.4°

**Table 4 sensors-24-04905-t004:** Root mean square error (RMSE) of the angles’ roll and pitch of the third flight.

Angle	1. RMSE IMU—Radar	2. RMSE IMU—Filtered Radar	3. RMSE IMU—Filtered Radar Main Flight Parts
Roll	5.6°	5.5°	5.1°
Pitch	8.7°	8.0°	7.8°

## Data Availability

The data used to support the findings of this study are available from the corresponding author upon request.

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
