# Peer review of "Flight Attitude Estimation with Radar for Remote Sensing Applications"

_sensors, 2024, doi:10.3390/s24154905_

Round 1

Reviewer 1 Report (Previous Reviewer 1)

Comments and Suggestions for Authors

This paper introduces a method to estimate the attitude of unmanned aerial vehicle using radar data. The proposed method is proven to be feasible according to experiment results.

Author Response

Dear reviewer 1, thank you very much for your review!

Reviewer 2 Report (Previous Reviewer 2)

Comments and Suggestions for Authors

The authors have provided point-by-point responses to the comments. I would like to recommend the authors to add following references in the text.

1. Tracking Multiple Autonomous Ground Vehicles Using Motion Capture System Operating in a Wireless Network

Author Response

Dear reviewer 2, thank you for the comment. We revised the introduction and added the topic tracking of UAV´s with motion capture systems. In addition to the suggested reference, we included two more.

“In order to identify drones, especially in restricted airspaces, motion capture systems are used to detect and track UAVs [13], [14], [15].”

[13]       S. A. Memon et al., “Tracking Multiple Autonomous Ground Vehicles Using Motion Capture System Operating in a Wireless Network,” IEEE Access, vol. 12, pp. 61780–61794, 2024, doi: 10.1109/ACCESS.2024.3394536.

[14]       S. A. Memon and I. Ullah, “Detection and tracking of the trajectories of dynamic UAVs in restricted and cluttered environment,” Expert Systems with Applications, vol. 183, p. 115309, Nov. 2021, doi: 10.1016/j.eswa.2021.115309.

[15]       S. A. Memon, H. Son, W.-G. Kim, A. M. Khan, M. Shahzad, and U. Khan, “Tracking Multiple Unmanned Aerial Vehicles through Occlusion in Low-Altitude Airspace,” Drones, vol. 7, no. 4, Art. no. 4, Apr. 2023, doi: 10.3390/drones7040241.

This manuscript is a resubmission of an earlier submission. The following is a list of the peer review reports and author responses from that submission.

Round 1

Reviewer 1 Report

Comments and Suggestions for Authors

Unfortunately this paper is poorly written and it should be thoroughly revised before it can be considered for publication.

First of all, in the title/abstract/introduction/conclusion, the authors repeatedly mention ‘position estimation’, but the paper’s main content (especially the experiment results) is nothing but the UAV’s pitch and roll angles. Maybe the authors are actually talking about the UAV’s attitude rather than its position?

Secondly, no matter what the authors are actually considering (either position or attitude), they should first give detailed measurement model of the sensors (such as the radar and the IMU), but the authors fail to give any meaningful descriptions, i.e. the whole theoretic part of this paper is useless.

Thirdly, assuming that we are discussing UAV’s pitch and roll, a reliable (i.e. with high enough precision) attitude reference should be available for the experiments, so as to quantitatively evaluate your proposed method’s performance (e.g. giving the root mean square errors of pitch and roll, rather than drawing some curves only).

Reviewer 2 Report

Comments and Suggestions for Authors

Please see attached file. thanks

Comments on the Quality of English Language

 Extensive editing of English language required.